# Enhancing Bitcoin Price Volatility Estimator Predictions: A Four-Step Methodological Approach Utilizing Elastic Net Regression

Georgia Zournatzidou[1], Ioannis Mallidis[2], Dimitris Farazakis[*3], and Christos Floros[4]

[1]Researcher at University of Western Macedonia Kozáni, Greece
[2]Department of Statistical and Insurance Science, University of Western Macedonia, 50100 Kozani, Greece
[3]Department of International and European Economic Studies, School of Economics, University of Western Macedonia & Institute of Applied and Computational Mathematics, FORTH, Heraklion, Greece
[4]Department of Accounting and Finance, Hellenic Mediterranean University, 71410 Heraklion, Greece
{zournatzidou.georgia@gmail.com, imallidis@uowm.gr, dfarazakis@uowm.gr, cfloros@hmu.gr}

## Abstract

This paper, [1], provides a computationally efficient and novel four-step methodological approach for predicting volatility estimators derived from bitcoin prices. Firstly, open, high, low, and close bitcoin prices are transformed into volatility estimators using Brownian motion assumptions and logarithmic transformations. Secondly, the optimal number of time-series lags are extracted by the Random Forest Regression (RFR) using the Mean Decrease in Impurity (MDI) criterion. Thirdly, we apply the Elastic Net Regression (ENR) to the volatility estimator's dataset, while the final fourth step assesses the predictive accuracy of ENR, compared to Decision Tree Regression (DTR), RFR, and Support Vector Regression (SVR).

## 1 Introduction

In the rapidly evolving landscape of financial markets, the accurate prediction of cryptocurrency prices is critical for a wide range of financial stakeholders, due to their highly volatile behaviour. Volatility estimators, which are critical tools necessary for analysing cryptocurrency price movements, constitute a key component used to quantify the intrinsic uncertainty and risk present in the market [2]. ENR stands out as a highly effective tool for addressing the challenges emerging from financial time series analysis. With respect to the stochastic behaviour of financial data and through its $L_2$ penalty, ENR stabilizes the model's predictions by reducing the magnitude of the impacts of correlated predictors, which is particularly useful when data encompass extreme variations [3]. Additionally, by penalizing the size of the coefficients, ENR minimizes the impact of outliers that could affect predictions in more simplified regression models.

---

*Corresponding Author.

## 2 Methodology

Four-step methodological approach **Step 1:** Deriving the functions quantifying the volatility estimators of open, high, low, and close cryptocurrency prices as the Parkinson Volatility Estimator, [4], with equation

$$\hat{\sigma_p^2} = \frac{(p_{max} - p_{min})^2}{4 \ln 2},$$

where $p_{max} = \ln(High) - \ln(Open)$, $p_{min} = \ln(Low) - \ln(Open)$. Similarly, we get the estimators of the Garman–Klass Estimator, [5], the Rogers–Satchell estimator, [6], and the estimator similar to [7]. **Step 2:** Then, we employ the RFR model to optimize the selection of time-series lags in our predictive model. The approach begins by constructing a forest of decision trees, where each tree is built from a bootstrap sample of the data. The RFR model utilizes the MDI criterion to evaluate the importance of each lag in predicting the volatility estimator [8]. The optimal number of lags are the lags resulting to an 85% cumulative importance. This criterion has been empirically selected as it achieves a balance between model complexity and explanatory power, ensuring that the model retains the most significant predictors while avoiding overfitting by excluding less impactful variables. **Step 3:** Having selected the optimal number of lags, we finalized the dataset structure and split the dataset to an 80% train and 20% test set. We then formulated the ENR model and fit the model on the train set. The ENR model constitutes a combined statistical tool of Ridge and Lasso (Least Absolute Shrinkage and Selection Operator) regression, based on the OLS method. Ridge regression, Lasso regression, and ENR are techniques used in the field of machine learning and statistics for regularization, which helps in reducing model complexity and preventing overfitting. The algorithm is expressed as

follows,

$$(\hat{\beta}_0, \hat{\beta}_j)_{EN} = argmin_{(\hat{\beta}_0, \hat{\beta}_j)} \Big[ \sum_{t=1}^{T} (y_t - \hat{y}_t)^2$$
$$+ \frac{\lambda(1-\alpha)}{2} \sum_{j=1}^{J} \beta_j^2 + \lambda\alpha \sum_{j=1}^{J} |\beta_j| \Big]$$

for $(\hat{\beta}_0, \hat{\beta}_j) \in R^{J+1}$. In order to optimize the selection of the ENR's hyperparameters, $\alpha$ and $\lambda$, which dictate the balance between the $L_2$ and $L_1$ penalties as follows

$$\|\beta_j\|_2^2 = \sum_{j=1}^{J} \beta_j^2, \quad \|\beta_j\|_1 = \sum_{j=1}^{J} |\beta_j|.$$

Here, $\beta_0$ expresses the intercept, $\beta_j$ the vector of the independent variable coefficients, and $y_t, \hat{y}_t$ the actual and predicted values of the prices at time t, respectively. T is the number of dataset periods (train or test) and $\lambda$ is the regularization parameter. The parameter $\alpha$ balances the Lasso and Ridge regression components.

We employ the RandomizedSearchCV method using the Python programming language on the train set. This methodological approach is implemented using libraries such as scikit-learn for the RandomizedSearchCV functionality and NumPy for numerical operations. We randomly select combinations from a predefined grid of $\alpha$ and $\lambda$ values and evaluate the performance of each combination using k-fold cross-validation. This stochastic sampling method is not only computationally less intensive than an exhaustive grid search, but also provides a practical compromise between thoroughness and efficiency. **Step 4:** To this end and for providing a comparative assessment of the ENR's predictive accuracy, we compare ENR with traditional machine learning regression models such as the DTR, the RFR, and the SVR. Considering the optimally selected number of lags for each volatility estimator, these models are fitted on the 80% train dataset for determining their optimal parameter values and employed on predicting the 20% test set dependent variable value.

## 3 Numerical Results

Having determined the optimal number of lags using the RFR model, we proceed with modelling each Bitcoin price volatility estimator using ENR. The model results indicate that the intercepts and coefficients vary considerably across different estimators, suggesting unique underlying dynamics in each case. For $VE_1$, the results show minimal coefficient shrinkage ($\lambda = 0.0511$) and a nearly pure ridge behavior ($\alpha \approx 0$), leading to a low Mean Absolute Error (MAE), indicative of a robust model fit. In contrast,

$VE_2$ adopts a more Lasso-like approach ($\alpha = 0.9053$) with a moderate $\lambda$ (0.0940), resulting in a higher MAE, which might imply less predictive accuracy. $VE_3$ exhibits the highest regularization ($\lambda = 0.1116$) and a balanced $\alpha$ (0.7369), yet it scores the highest MAE among the models, possibly indicating issues with model suitability or data fit. Finally, $VE_4$ demonstrates the highest degree of coefficient shrinkage ($\lambda = 0.5613$) with a predominance of ridge regression characteristics ($\alpha = 0.0762$), which yields a relatively low MAE. These results highlight the critical importance of tuning the $\lambda$ and $\alpha$ parameters in elastic net models to balance the trade-off between bias and variance effectively, particularly in financial datasets where volatility estimators can behave unpredictably. To further evaluate the predictive power of ENR compared to other regression models, we employed the same process for predicting the dependent target variable values of each bitcoin price volatility estimator considering DTR, RFR, and SVR.

| Model | $VE_1$ | $VE_2$ | $VE_3$ | $VE_4$ |
|-------|--------|--------|--------|--------|
| ENR | 0.0016 | 1.2366 | 1.2833 | 0.2076 |
| DTR | 0.0020 | 1.8767 | 2.0269 | 0.4224 |
| RFR | 0.0018 | 1.3853 | 1.5067 | 0.3359 |
| SVR | 0.0839 | 0.8495 | 0.9483 | 0.2492 |

The results reveal that different regression models exhibit varied effectiveness across the volatility estimators of Bitcoin prices, corresponding to open, high, low, and close values. ENR performs exceptionally well with open and close prices, where the predictive relationships may be more linear and less susceptible to sudden, non-linear shifts typically seen during trading hours. The regularization in ENR helps to prevent overfitting while effectively capturing the linear trends that might govern the open and close price movements. On the other hand, SVR shows superior performance in estimating the volatility of high and low prices as these prices often experience spikes and drops driven by transient news and intra-day market sentiments [9]. SVR, with its capability to handle non-linear relationships through kernel transformations, can capture these dynamics more accurately.

## 4 Conclusion

This research significantly contributes to the field of cryptocurrency analysis by highlighting the efficacy of different regression models, including ENR, in forecasting Bitcoin price volatility. Utilizing a robust dataset, this study emphasizes the importance of a methodologically sound approach to lag selection (Random Forest). Finally, the ENR model performed best for open and close prices.

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
