# OpenReview forum: "Enhancing Bitcoin Price Volatility Estimator Predictions: A Four-Step Methodological Approach Utilizing Elastic Net Regression"
_NLDL.org/2026/Abstracts_Track — NLDL 2026 Abstracts_

### Official Review · Reviewer_oB1K · 2025-11-02

**Soundness:** 3
**Correctness:** 3
**Rating:** 4
**Confidence:** 2

**Summary:**

The authors propose a 4-step methodological approach for volatility estimation of Bitcoin using Elastic Net Regression (ENR). The process begins by deriving multiple volatility estimators from open, high, low, and close prices based on Brownian motion assumptions. Next, the optimal number of time-series lags is selected using Random Forest Regression. ENR is then applied to model the volatility estimators, with hyperparameters tuned through RandomizedSearchCV. Finally, the model’s predictive performance is compared against Decision Tree Regression (DTR), Random Forest Regression (RFR), and Support Vector Regression (SVR). Results show that the authors approach provides superior predictive accuracy for open and close prices, while SVR performs better for high and low prices.

**Strengths:**

- Volatility prediction of cryptocurrencies like Bitcoin is a relevant financial topic with clear practical applicability.
- The idea to use ENR, which is less sensitive to outliers and extreme variations than competing methods, is technically sound for cryptocurrencies, which are know to be highly unpredictive.
- The four-step approach of volatility estimation, lag selection, model optimization and evaluation proposed by the authors seems technically, and is easy to follow.
- The authors ENR approach achieves best estimates of open and closed prices compared with competing methods.

**Weaknesses:**

- The abstract lacks a section describing related work, making it difficult for the reader to position the authors contribution in relation to the research field.
- The writing is not very accessible for readers unfamiliar with the field, and there are missing explanations for acronyms such as OLS, $V$ $E_1$ to $V$ $E_4$  in the abstract. There are also no figures to supplement the explanations of the proposed method.

---

### Official Review · Reviewer_z5Bh · 2025-11-03

**Soundness:** 3
**Correctness:** 3
**Rating:** 2
**Confidence:** 3

**Summary:**

This work proposes a computationally efficient four-step approach for predicting Bitcoin price volatility. First, volatility estimators are derived from open, high, low, and close prices using Brownian motion and logarithmic transformations. Next, the optimal time-series lags are selected via Random Forest Regression (RFR) using the Mean Decrease in Impurity criterion. The Elastic Net Regression (ENR) model is then trained, and its predictive performance is compared against Decision Tree, Random Forest, and Support Vector Regression models.

**Strengths:**

* Well written
* Clear Experimental setup

**Weaknesses:**

* The main concern is that, according to the NLDL call for abstracts, all submissions should undergo single-blind peer review, which does not seem to have been followed in this case.

---

### Decision · Program_Chairs · 2025-11-05

**Decision:**

Accept

**Comment:**

The reviewers found the abstract borderline, yet the PCs believe it will be of interest to the community and should have the opportunity be presented.